# Ultrastructure and Spectral Characteristics of the Compound Eye of *Asiophrida xanthospilota* (Baly, 1881) (Coleoptera, Chrysomelidae)

**DOI:** 10.3390/insects15070532

**Published:** 2024-07-13

**Authors:** Zu-Long Liang, Tian-Hao Zhang, Jacob Muinde, Wei-Li Fan, Ze-Qun Dong, Feng-Ming Wu, Zheng-Zhong Huang, Si-Qin Ge

**Affiliations:** 1Key Laboratory of Zoological Systematics and Evolution, Institute of Zoology, Chinese Academy of Sciences, Beijing 100101, China; liangzulong@ioz.ac.cn (Z.-L.L.); shanezth@126.com (T.-H.Z.); mulwa.muinde@ioz.ac.cn (J.M.); 15110637937@163.com (W.-L.F.); ncuskdongzequn@163.com (Z.-Q.D.); huangzz@ioz.ac.cn (Z.-Z.H.); 2University of Chinese Academy of Sciences, Beijing 100049, China

**Keywords:** ultrastructure, compound eye, electroretinogram, phototaxis, insect vision, *Asiophrida xanthospilota*

## Abstract

**Simple Summary:**

The flea beetle *Asiophrida xanthospilota* (Baly, 1881) is a serious forest pest specifically damaging the common smoketree *Cotinus coggygria*. To understand how this beetle sees the world, we used scanning electron microscopy, transmission electron microscopy, micro-computed tomography, and three-dimensional reconstruction to investigate the external morphology and internal ultrastructure of the compound eye. The examination showed that of *Asi. xanthospilota* has apposition eye like other leaf beetles, consisting of a subplano-convex cornea, an acone of four cone cells, eight retinular cells along with an open rhabdom, as well as two primary pigment cells and about 23 secondary pigment cells. Interestingly, retinular cell 8 does not contribute to the rhabdom. We also investigated the spectral sensitivity by electroretinogram tests and phototropism experiments. Electroretinogram tests showed that *Asi*. *xanthospilota* exhibits the strongest sensitivity to blue and green lights but the weakest to red light. Phototropism experiments also revealed that this beetle has the strongest reaction to blue light.

**Abstract:**

In this study, the morphology and ultrastructure of the compound eye of *Asi. xanthospilota* were examined by using scanning electron microscopy (SEM), transmission electron microscopy (TEM), micro-computed tomography (μCT), and 3D reconstruction. Spectral sensitivity was investigated by electroretinogram (ERG) tests and phototropism experiments. The compound eye of *Asi. xanthospilota* is of the apposition type, consisting of 611.00 ± 17.53 ommatidia in males and 634.8 0 ± 24.73 ommatidia in females. Each ommatidium is composed of a subplano-convex cornea, an acone consisting of four cone cells, eight retinular cells along with the rhabdom, two primary pigment cells, and about 23 secondary pigment cells. The open type of rhabdom in *Asi. xanthospilota* consists of six peripheral rhabdomeres contributed by the six peripheral retinular cells (R1~R6) and two distally attached rhabdomeric segments generated solely by R7, while R8 do not contribute to the rhabdom. The orientation of microvilli indicates that *Asi. xanthospilota* is unlikely to be a polarization-sensitive species. ERG testing showed that both males and females reacted to stimuli from red, yellow, green, blue, and ultraviolet light. Both males and females exhibited strong responses to blue and green light but weak responses to red light. The phototropism experiments showed that both males and females exhibited positive phototaxis to all five lights, with blue light significantly stronger than the others.

## 1. Introduction

Compound eyes serve as the primary photoreceptive organs of adult insects, fulfilling crucial roles in recognizing conspecifics, distinguishing and evading predators, searching for food, and attacking prey, as well as navigating and orientating during walking, swimming, and flying [1]. Despite their size and shape, the compound eyes of insects are composed of clusters of repeated basic units called ommatidia. Each ommatidium can be divided into three components: (1) a dioptric apparatus consisting of a corneal lens and a crystalline cone; (2) a photoreceptive element (photoreceptor) consisting of retinular cells along with the rhabdom formed by these cells; and (3) a light-insulating element consisting of primary and secondary pigment cells [2].

According to the optical design, there are four basic types of compound eyes in insects: apposition eyes, hybrid compound/camera eyes, neural superposition eyes, and optical superposition eyes [3]. Compound eyes of coleopterans are mainly of the apposition and optical superposition types, with hybrid eyes occurring only in groups of Strepsiptera and neural superposition eyes occurring only in groups of Diptera. In the compound eyes of the apposition type, each ommatidium is an independent imaging unit that is completely optically isolated from others. This type of compound eyes typically produces a clear image but often sacrifices photosensitivity. Thus, apposition eyes are primarily utilized by diurnal insects that are active during well-lit hours, such as dragonflies [4], butterflies [5], bees [6], and some beetles [7]. In superposition eyes, on the other hand, the dioptric apparatus and photoreceptor are separated by a pigment-free region known as the “clear zone” [8]. Thus, the rhabdom of an ommatidia can not only receive light from its own lens but also allow light from neighbouring ommatidia to pass through the clear zone. This design enables the concentration of light coming from multiple facets, thereby improving photosensitivity in dim-light environment. Superposition eyes are typically found in insects that are considered crepuscular or nocturnal [9], such as moths and some beetles [10,11]. The compound eyes of leaf beetles (family Chrysomelidae) species are of the apposition type, as indicated by previous studies [12,13,14], consistent with their diurnal activity [15].

The ultrastructure of insect ommatidia has long been studied using conventional imaging techniques such as light and electron microscopy [16]. However, these methods normally provide only two-dimensional image data. Micro-CT offers an alternative for noninvasive 3D imaging and has been used in some studies to investigate the morphology and structure of insect compound eyes [13,14,17,18,19]. In recent years, high-resolution 3D imaging techniques such as synchrotron source X-ray micro-computed tomography (syn-μCT) and nanotomography (nano-CT) for compound eye morphology have been developed [16,20], indicating that high-resolution imaging has great potential for the study of insect compound eye morphology. In addition, other studies have developed methods and algorithms to study the optical parameters of compound eyes using micro-CT image data [21,22].

Animals are capable of detecting a wide range of light spectra to extract colour information. Our current knowledge of colour vision of Chrysomelidae is still limited. Electrophysiological analysis revealed that the Colorado potato beetle, *Leptinotarsa decemlineata* (Chrysomelinae), has three sensitivity peaks at UV, blue, and green wavelengths [23]. Similarly, *Callosobruchus maculatus* (Bruchinae) and *Agasicles hygrophila* (Galerucinae, Alticini) also have positive responses to UV, blue, and green stimuli [13,14]. Notably, *Aga. hygrophila* exhibits strong responses to yellow and red stimuli, even stronger than those to green and blue lights [14]. Yet, it is not clear what may cause the strong sensitivity to long-wavelength lights.

The flea beetle *Asiophrida xanthospilota* (Baly, 1881) (Coleoptera, Chrysomelidae, Galerucinae, Alticini) is a serious forest pest that specifically damages *Cotinus coggygria* Scopoli, a famous red leaf landscape tree in northern China [15,24]. The larvae of *Asi. xanthospilota* feed on flower buds and young leaves of *Cog. coggygria*, while the adults feed on mature leaves of the same species, causing breaches or holes in the leaves and even leading to the death of trees during outbreaks [25,26]. Some studies suggested that visual cues play a crucial role in host- and mate-finding processes of leaf beetles [27,28]. Thus, understanding the mechanisms of visual perception in *Asi. xanthospilota* may help us develop more effective methods to control this species. To improve our understanding of the visual system of *Asi. xanthospilota*, we examine its external morphology and internal ultrastructure using scanning electron microscopy, transmission electron microscopy, and micro-computed tomography (CT). Additionally, we investigate spectral sensitivity through electroretinogram tests and phototropism experiments.

## 2. Materials and Methods

### 2.1. Materials

Individuals of adult *Asi. xanthospilota* were collected from the China National Botanical Garden in June and July (Figure 1). In the laboratory, the beetles were raised in transparent plastic boxes and fed with fresh leaves of *Cotinus coggygria* collected from the China National Botanical Garden (40.00° N, 116.20° E).

### 2.2. Scanning Electron Microscopy (SEM)

The heads of the beetles were separated from the body for SEM examination, and the antennae were removed. The heads were then cleaned three times with 75% ethanol. The samples were subsequently dehydrated in a graded series of ethanol with concentrations of 75%, 80%, 85%, 90%, and 95%, each maintained for 30 min, followed by three additional treatments in pure ethanol. The dehydrated samples were dried using a critical point dryer (Leica EM CPD 300, IZCAS, Beijing, China) for 29 cycles. After being mounted on a rotatable specimen holder, the samples were sputter-coated with gold for 120 s using the Sputter Coater (Leica EM SCD050, IZCAS, Beijing, China). The scanning electron microscope (ESEM FEI Quanta 450, IZCAS, Beijing, China) was used to capture SEM images. Micrographs were captured at an accelerating voltage of 12.5 kV.

### 2.3. Transmission Electron Microscopy (TEM)

The heads of the specimens were fixed with a fixation solution containing 2.5% (*v*/*v*) glutaraldehyde (SPI, Inc., 111-30-8, Structure Probe, Inc., West Chester, PA, USA) and 4% paraformaldehyde in phosphate buffer (PB) (0.1 M, pH 7.4). Then, the samples were fixed with 2.5% (*v*/*v*) glutaraldehyde and 1% tannic acid in phosphate buffer, washed twice in PB and twice in ddH_2_O. Subsequently, the fixed samples were immersed in a 1% (*w*/*v*) OsO4 solution (TED PELLA, Inc., 18456, Ted Pella, Inc., Altadena, USA) and a 1.5% (*w*/*v*) potassium ferricyanide aqueous solution at 4 °C for 2 h. After being washed with ddH_2_O, the samples were dehydrated in a graded series of ethanol with concentrations of 30%, 50%, 70%, 80%, 90%, each for 10 min, followed by two treatments in 100% ethanol for 10 min and then twice in pure acetone for 10 min each. Subsequently, the samples were infiltrated in graded mixtures (8:1, 5:1, 3:1, 1:1, 1:3, 1:5) of acetone and Spurr’s resin (10 g of ERL 4221 (SPI, Inc., 02815), 8 g of DER 736 (SPI, Inc., 02830), 25 g of NSA (SPI, Inc., 02829), and 0.7% DMAE (SPI, Inc., Z02824)), and then pure resin. Finally, the samples were embedded in pure resin and polymerised for 12 h at 45 °C and 48 h at 70 °C. The ultrathin sections (70 nm thick) were obtained using a microtome (Leica EM UC6, IBPCAS, Beijing, China), double-stained with uranyl acetate and lead citrate, and examined using a transmission electron microscope (FEI Tencai Spirit 120 kV, IBPCAS, Beijing, China). Micrographs were captured at an accelerating voltage of 100 kV.

### 2.4. Micro-Computed Tomography and 3D Reconstruction

One female specimen of *Asi. xanthospilota* was used for micro-computed tomography and 3D reconstruction. The head and prothorax of the beetle were separated from the body, and the antennae were removed using tweezers. The sample was fixed in 75% ethanol for a day, and then dehydrated in a series of graded ethanol (75%, 80%, 85%, 90%, 95%, and thrice in 100%, each for 30 min). Then, the sample was dried in a freeze-dryer (Marin Christ, IZCAS, Beijing, China) for 12 h. It was then mounted on an Eppendorf tube and scanned using an X-radia scanner (Leica Micro XCT-400, IZCAS, Beijing, China) at a magnification of 4×. Images were captured at intervals of 5.5 s. 2D image stack obtained through micro-CT scanning was reconstructed (image size: 1012 × 1012 pixels, pixel size: 2.126 μm), and different compound eye structures were segmented using Amira software version 6.0.1. The segmented materials were imported into VG Studio Max 3.4.1 for rendering and visualization.

The eye radius (*r*) and the interommatidial angle (Δ*φ*) can be determined following the formulas below (Figure 2) [29]:$$r=\frac{{\left(s/2\right)}^{2}+h^{2}}{2h}$$
and
Δφ=d/r
where *s* refers to the length of the baseline of the eye; *h* refers to the longest distance from the curvature to the baseline; and *d* refers to the facet diameter of an ommatidium.

### 2.5. Electroretinogram

The Electroretinogram (ERG) testing involved the selection of healthy adult individuals, both females and males. After cryo-anaesthesia, a precise incision was made through the head and prothorax, resulting in the removal of antennae and prolegs. The glass electrodes, fabricated using a micropipette puller, were filled with conductive fluid (128.34 mM of NaCl, 4.69 mM of KCl, and 1.89 mM of CaCl_2_·2H_2_O in water). The reference electrode was inserted into the tissue of the prothorax, while the recording electrode was positioned on the surface of the compound eye. After stabilizing the potential signal, the light was activated to induce eye stimulation, and the active potential signal was recorded. The stimulation was conducted using LED beads emitting five different lights (red, yellow, green, blue, ultraviolet) with an approximate illuminance of ca. 100 lux. The amplified signal was captured by a computer using WinWCP: Strathclyde Electrophysiology Software v. 5.1.1.1.

### 2.6. Phototaxis Test

The phototaxis test was conducted using the L-shaped test chamber, which was modified from the design of previous studies [14,30]. Before each test, the samples were subjected to a 20 min dark adaptation period in the starting area (SA) prior to light stimulation. Subsequently, the light was activated, and the beetles were exposed to it for a duration of 5 min. Finally, the number of individuals present in both the light area (LA) and dark area (DA), as well as those remaining in the SA, was recorded. After each test, the inner side of the chamber was cleaned using 95% ethanol to prepare for the subsequent test. LED beads emitting red (620–625 nm), yellow (588–590 nm), green (515–525 nm), blue (460–465 nm), and ultraviolet (365–400 nm) wavelengths were used as light sources for phototaxis testing, with an approximate illuminance of ca. 100 lux, while a control test was conducted with no light at both ends of the chamber. For each light source set, nine replicate tests were performed for both males and females, with 10–20 samples used in each test. The phototactic response was calculated by the following formula:Positive phototaxis = (number of individuals in LA/total individuals) × 100%
Negative phototaxis = (number of individuals in DA/total individuals) × 100%
Non-phototaxis = (number of individuals in SA/total individuals) × 100%

### 2.7. Data Analysis

The numbers and areas of the ommatidia were calculated using ImageJ software v1.54j based on SEM images. A semi-schematic drawing of the ommatidium was created using Adobe Illustrator 2023 software.

ERG data were examined by Clampex software v. 10.6. Data of ERG and Phototaxis test were analyzed with SPSS 18.0 software and visualized using GraphPad Prism 7.00 software. Plates were created using Adobe Photoshop 2023 software.

The statistical significance of the differences between males and females was measured utilizing the unpaired two-tailed Student’s *t*-test. Comparisons between different lights were performed utilizing Tukey HSD one-way ANOVA analysis. A value of *p* < 0.05 was considered statistically significant (n.s., not significant; *, *p* < 0.05; **, *p* < 0.01; ***, *p* < 0.001). The *p*-value, standard error (SE), and number are indicated in each figure and legend.

## 3. Results

### 3.1. External Morphology

The compound eyes of both male and female *Asi. xanthospilota* exhibit similar external morphology, appearing ellipsoidal in shape and situated on the lateral sides of the head, protruding outwards (Figure 3a–d). Each eye contains an average of 611.00 ± 17.53 facets in males and 634.80 ± 24.73 in females, showing no significant difference (*p* > 0.05, n = 10) (Figure 3g). Yet, the surface area of the eyes is larger in females (338,787.50 ± 24,936.78 μm^2^) than in males (303,658.50 ± 17,451.61 μm^2^) (*p* < 0.05, n = 10) (Figure 3h). The surface of facets is smooth, with very few short interfacetal hairs scattering among them (Figure 3e). Most facets are regular hexagons (Figure 3c–e), while some are irregular pentagons, usually forming two adjacent short rows in some areas (Figure 3c,d,f). The area of each hexagonal facet (618.68 ± 49.40 μm^2^) is significantly larger than that of the pentagonal facet (527.31 ± 28.45 μm^2^) (*p* < 0.01, n = 20) (Figure 3i).

### 3.2. Internal Structures Organization

The length of the ommatidia is 163.98 ± 2.45 μm in males (n = 4) and 184.14 ± 6.80 μm in females (n = 4). Each ommatidium in *Asi. xanthospilota* consists of two distinct parts: a dioptric apparatus, which includes the cornea and crystalline cone, and photoreceptive elements composed of retinular cells along with rhabdom (Figure 4a,b). The primary pigment cells surround the proximal part of the cone, while the secondary pigment cells occupy the spaces between adjacent ommatidial units from the level of the cone to the basal membrane (Figure 4b and Figure 5i). The absence of the clear zone, with the cone cells in direct contact with the rhabdom, indicates that the compound eyes of *Asi. xanthospilota* conform to the apposition type.

#### 3.2.1. Dioptric Apparatus

The cornea is a subplano-convex lens, with the outer curve near plane and the inner curve much more convex (Figure 4c). The radius of curvature for the outer curve and inner curve are 35.03 ± 5.92 and 5.95 ± 1.56 μm, respectively (n = 3). The corneal thickness measures 44.61 ± 2.51 μm (n = 3). TEM micrographs of both cross sections and longitudinal sections show that the cornea has a laminated structure consisting of dense layers with alternating electron densities (Figure 4c and Figure 5a).

Four wedge-shaped cone cells lie below the corneal lens, each contributing one-quarter to the crystalline cone (Figure 4d and Figure 5b). The crystalline cone is of the acone type. The cone cells directly contact the cornea, with cytoplasm situated at the distal top and the core positioned below (Figure 4c). The large cone cell nuclei are tightly packed within the cones.

#### 3.2.2. Photoreceptive Elements

The photoreceptive layer of each ommatidium consists of six peripheral cells (R1–R6) and two central cells (R7–R8), with each cell extending throughout the entire length of the photoreceptive layer (Figure 4a,e). Each of the six peripheral cells gives rise to a rhabdomere on the inner axial side of the distal part, which connects to the neighbouring rhabdomere to form a ring-shaped peripheral part of the rhabdom (Figure 5e). The two central retinular cells, surrounded by peripheral cells, are unequal, together forming an approximately hexagonal outline in cross sections. The smaller R8 is dumbbell-like in cross sections, with its two ends connected by a narrow bridge restricted by rhabdomere 7 (Rh7). R7 generates two rhabomeric segments that form the central rhabdom, which are narrowly connected distally and extend half the length of the cell (Figure 5d,f,h). In contrast, R8 does not contribute to the rhabdom at all. The central rhabdomere is completely isolated from the peripheral part, confirming that *Asi. xanthospilota* has ommatidia with open rhabdom. The central part is longer than the peripheral part at both ends. The cell nucleus of each retinular cell lies below the rhabdomeres (Figure 4a,b and Figure 5g). The axons of all the retinular cells are arranged as a compact bundle at the proximal level, with one axon distinctly larger than the others, presumably the axon of R7 or R8. The axon bundle is surrounded by confusedly arranged secondary pigment cells above the basal membrane and then runs into the brain through the basal membrane (Figure 5i).

The rhabdomere is composed of a series of regularly aligned microvilli. The microvilli of peripheral rhabdomeres are roughly oriented towards the centre of the ommatidium. The microvilli of the central rhabdomere are perpendicular to the axis of R8 (Figure 5f). The orientation of the microvilli suggests that *Asi. xanthospilota* is unlikely to perceive polarized light.

### 3.3. 3D Reconstruction of the Compound Eye

After reconstruction, three distinct layers are observed in the reconstructed 3D images, including the cornea layer, the crystalline cone layer, and the photoreceptive layer (Figure 6). The cornea layer is the outermost part of the compound eye and is composed only of the cornea. Beneath the cornea layer lies the crystalline cone layer, which includes the crystalline cone, the primary pigment cells, and part of the secondary pigment cells at the same level. The delimitation between each structure is difficult to distinguish. The photoreceptive layer, consisting of retinular cells and surrounding secondary pigment cells, is located beneath the crystalline cone layer and connects to the brain at its proximal end.

The compound eyes have a large visible area on the dorsal, frontal, and lateral view (Figure 6a–c), indicating that the vision range of *Asi. xanthospilota* is primarily located on the top, front, and sides of the head. The spatial resolution of the eyes can be estimated using the interommatidial angle (Δ*φ*) (for Formulas, see Section 2.4). Parameters *s*, *h*, and *d* measure 796 μm, 286 μm, and 26.77 μm, respectively. According to the formulas, the eye radius (*r*) and interommatidial angle (Δ*φ*) of the eyes of *Asi. xanthospilota* are calculated as 419.93 μm and 0.064 rad, respectively.

### 3.4. ERG Testing

In ERG testing, both female and male individuals exhibited responses in their compound eyes to all five light wavelengths (Figure 7). Among females, the green-light stimulation elicits the highest signal intensity, which is significantly greater than that of other light stimulations except for blue light (Figure 8a). Conversely, red-light stimulation results in the lowest signal intensity, significantly lower than that of all other wavelengths (Figure 8a). In males, in contrast, blue-light stimulation elicits the highest signal intensity, but there is no significant difference observed among blue-, green-, and UV-light stimulations (Figure 8b). Moreover, the response to red-light stimulation in males is significantly lower compared to that of all other light sources (Figure 8b). When comparing the response intensity of red and yellow light wavelengths between males and females, no significant difference was found. However, in males, the response intensity to green-, blue-, and UV-light stimulation is significantly higher than in females.

### 3.5. Phototaxis Testing

The results of behaviour experiments indicate that all five light wavelengths exhibit significantly higher rates of positive phototaxis compared to negative phototaxis in both males and females (Figure 9a,b). Furthermore, the positive phototaxis rates of all five wavelengths exhibit a statistically significant increase compared to the control group in both sexes (Figure 9c,d). The results of our experiments thus demonstrate that *Asi. xanthospilota* exhibits phototaxis towards these light stimuli. Both males and females exhibit the highest phototaxis towards blue light wavelength, showing a significant difference compared to other light stimuli. No significant difference in phototaxis is observed among UV, green, yellow, and red light wavelength (Figure 9c,d).

## 4. Discussion

### 4.1. Ultrastructure of Ommatidia

The ommatidia structure is highly conserved across different lineages of Chrysomeloidea. The ommatidia of Chrysomeloidea species exhibit a consistent structural organization, characterized by the presence of an apposition eye with acone and open rhabdom [10,12]. Our study revealed that the compound eyes of *Asi. xanthospilota* are consistent with the general structural design, with each ommatidium consisting of a subplano-convex lens, an acone consisting of four cone cells, two primary pigment cells, about 23 secondary pigment cells, and 8 retinular cells.

The eye of *Asi. xanthospilota* is of acone type, suggesting that the cone has weak light-gathering capacity. Thus, the ommatidium relies solely on the cornea to refract the light passing through the facet. The low refractive index of acone and subplano-convex design of cornea indicates that the refractive function is largely executed by its strongly convex inner surface [10]. The photoreceptive elements of *Asi. xanthospilota* show an open rhabdom composed of six peripheral rhabdomeres and two distally attached rhabdomeric segments solely generated by R7, while R8 does not contribute to the formation of the rhabdom at all. The presence of an open rhabdom eye is considered a synapomorphy of the Cucujiformia [12], which has been supported by most available evidence (Appendix A) [7,13,14,31,32,33]. Although most species follow a similar pattern, the organization of the rhabdomeres varies among different lineages in Chrysomelidae and Cerambycidae.

In most species, the peripheral rhabdomeres are arranged in a ring in cross sections. However, in some long-horn beetles (including species of Lepturinae and Lamiinae), the peripheral rhabdomeres have no lateral contact with neighbouring units. Additionally, the peripheral rhabdomeres are normally shorter than the central ones. In some extreme cases, such as in *Stenopterus ater*, the length of peripheral rhabdomeres is greatly reduced or they may even be completely missing in certain ommatidia [34]. The peripheral rhabdomeres of *Asi. xanthospilota* follow a typical organization, forming a hexagonal ring that encircles the longer central rhabdomere.

The central rhabdomeres exhibit stronger variation compared to the peripheral ones. Wachmann [35] proposed two basic patterns of open rhabdoms based on the relative position of the central rhabdom, which were termed “Grundmuster 1 and 2”. In Grundmuster 1, the central rhabdomeres are completely isolated from peripheral rhabdomeres, whereas in Grundmuster 2, the central rhabdomeres are partially attached to or fused with peripheral rhabdomeres. Our results indicate that the rhabdomere organization of *Asi. xanthospilota* belongs to Grundmuster 1, similar to most Chrysomelidae species except those from the subfamily Chrysomelinae (Appendix A) [12,13,14]. The separation of central rhabdomeres from peripheral rhabdomeres is considered a design for enhancing spatial resolution [34].

Among the compound eyes of Grundmuster 1, one of the basic arrangement modes of central rhabdomeres is the fusion of Rh7 and Rh8 to form an oval central rhabdom, equally contributed by two central retinular cells and usually having microvilli oriented in the same direction. This arrangement mode is common in various lineages of Cucujiformia, including Cleridae, Cerambycidae (Lepturinae, Cerambycinae, Lamiinae), and Chrysomelidae (Donacinae, Criocerinae, Eumolpinae, Cassidinae, Cryptocephalinae and Bruchinae), which may represent the ancestral state of rhabdomere arrangement (Appendix A) [12,13,34]. Some other species, by contrast, are characterized by unequal central rhabdom. The central rhabdom of *Orsodacne ceras* (Orsodacninae), *Zeugophora flavicollis* (Megalopodinae), and *Agelastica alni* (Galerucinae) is comprised of three parts: two contributed by the dominant central retinular cell (R7 to our understanding) and one by the other cell [12]. A similar arrangement of rhabdomeres is found in *Aga. hygrophila* (Galerucinae, Alticini) [14]. The central rhabdomere arrangement in *Asi. xanthospilota is* most similar to that of *Aga. hygrophila*. Both species have R7 contributing equally to two parts of rhabdomeres, but the rhabdomere of R8 is completely absent in *Asi. xanthospilota*. The complete absence of the rhabdomere from R8 has never been reported before in beetles with open rhabdoms. It is unclear how this may affect the vision of the beetle and what role R8 without rhabdomere may play in the visual system.

Some species of Cerambycidae and Chrysomelidae, such as *Monochamus alternatus*, *Dorcatypus tristis*, *Donacia simplex*, and *Melasoma aenea* (Appendix A), have microvilli oriented in two perpendicular directions, which is typically considered a design for polarization sensitivity [7,12,34]. In contrast, like most other species of Chrysomelidae, the microvilli in different rhabdomeres of *Asi. xanthospilota* roughly orient themselves in three directions, indicating that it is unlikely to be a polarization-sensitive species.

### 4.2. Spectral Sensitivity

Insects generally possess three opsin proteins (UV, SW, and LW), which typically have maximal sensitivity to ultraviolet (~350 nm), blue (~440 nm), and green (~530 nm) light, respectively, resulting in trichromatic visual systems [36]. Some insects expand their sensitive spectral ranges into the violet and red regions through gene duplications of the SW and LW opsins [37,38,39]. Some molecular studies have revealed that the SW opsin class, which is typically sensitive to blue light, has been lost in some beetle lineages, including Cicindelidae [40], Scarabaeidae [40], Dytiscidae [41,42], Buprestidae [43], Lampyridae [44,45], and Tenebrionidae [46]. However, electrophysiological evidence indicates the presence of blue photoreceptors in the compound eyes of some beetles, such as Coccinellidae [40,47], Chrysomelidae [23], and Buprestidae [48]. Recent transcriptomic analysis of opsin genes across beetle and relative lineages has revealed that the SW opsin class was lost before the origin of modern beetles, leading to the absence of visual sensitivity to blue wavelengths in the ancestor of beetles. However, subsequent gene duplications of UV opsins have independently occurred over 10 times, leading to the restoration of blue sensitivity in Coccinellidae, Chrysomelidae, and Buprestidae [36].

According to electrophysiological and molecular studies, Chrysomelidae normally have three types of photoreceptors, each with maximal sensitivity to ultraviolet, blue, and green light, respectively [23,36]. Our ERG testing shows that *Asi. xanthospilota* exhibits strong responses to ultraviolet, blue, and green lights, indicating the presence of these three types of photoreceptors in its compound eyes. Additionally, individuals of *Asi. xanthospilota* exhibit weak responses to yellow and red light, probably due to the response of green-sensitive photoreceptors. In contrast, ERG testing on another flea beetle species, *Aga. hygrophila*, revealed that this species exhibits strong responses to yellow and red light, which are significantly stronger than its responses to blue and even green lights [14], indicating the presence of a photoreceptor with maximal sensitivity in the yellow-to-red spectrum. Red-sensitive photoreceptors are very rare in the visual system of beetles, though they have been documented in a few species, including *Carabus nemoralis* and *Car. auratus* (Carabidae) [49], *Agrilus planipennis* (Buprestidae) [48], and *Pygopleurus israelitus* (Glaphyridae) [50]. The distant phylogenetic placement of Carabidae, Buprestidae, and Glaphyridae suggests that red-sensitive photoreceptors may have independently evolved multiple times within Coleoptera [50]. The actual mechanism leading to the existing red-sensitive photoreceptors is not yet clear, but it may be caused by some form of spectral filtering [50] or duplication of LW opsin [36]. Unfortunately, current studies are insufficient to draw an unambiguous conclusion about the specific photoreceptors possessed by *Asi. xanthospilota* and *Aga. hygrophila*. Further intracellular electrophysiological and transcriptomic analyses are required for verification.

Photoreceptors typically express only one visual pigment per cell [51], though a few exceptions have been reported [46,52,53]. In insects with open rhabdom eyes, such as Diptera, Lepidoptera, Coleoptera, and Hymenoptera, LW opsin is consistently expressed by peripheral photoreceptors (R1–R6). However, central rhabdomeres (R7, R7-like, and R8) may express UV opsin, SW opsin, as well as LW opsin [1]. In situ hybridization analysis revealed that in darkling beetles (Tenebrionidae), R7 co-expresses UV opsin and LW opsin across the entire retina, while the other seven photoreceptors express exclusively LW opsin [46,54]. In Coccinellidae, R7 and R8 are identified as UV and blue photoreceptors in *Coccinella septempunctata* [47]. According to the result of Sharkey et al. [36], the blue sensitivity of *Coc. septempunctata* is due to duplications of UV opsins, as the SW opsins are lost in beetles. Considering the loss of rhabdomere formation in the R8 in *Asi. xanthospilota*, we assume that both UV opsin and its duplication are likely expressed within rhabdomere R7, enabling it to detect both UV and blue light. Rh7 of *Asi. xanthospilota* is divided into two separate segments. It is an interesting question whether the two opsins co-occur and co-express in the entire retina or independently express in different segments. Moreover, *Asi. xanthospilota* has a very similar ommatidial structure with *Aga. hygrophila*, with the major exception being the presence of Rh8 in *Aga. hygrophila* [14]. Yet, compared to *Asi. xanthospilota*, *Aga. hygrophila* exhibits strong responses to red and yellow light. Thus, Rh8 is likely responsible for the sensitivity to red and yellow spectra, probably due to the gene duplication of LW opsin in Rh8. However, these hypotheses need to be verified through further investigation with more solid evidence.

## Figures and Tables

**Figure 1 insects-15-00532-f001:**
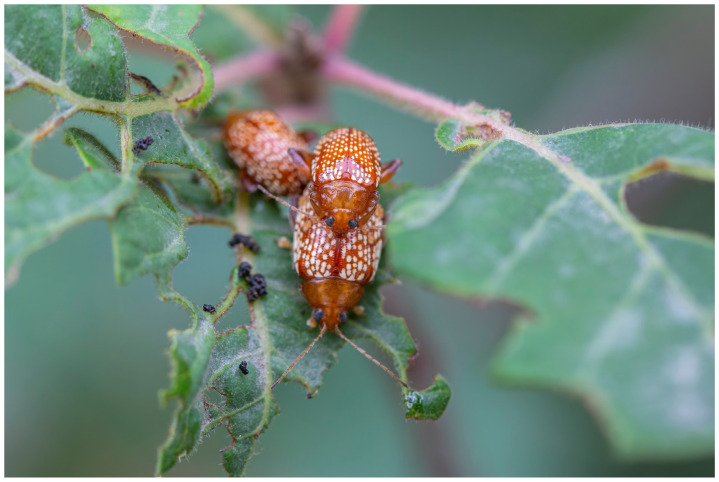
Adult *Asi. xanthospilota* are mating on host plant *Cotinus coggygria*. Photo taken by Dr. Zhengzhong Huang.

**Figure 2 insects-15-00532-f002:**
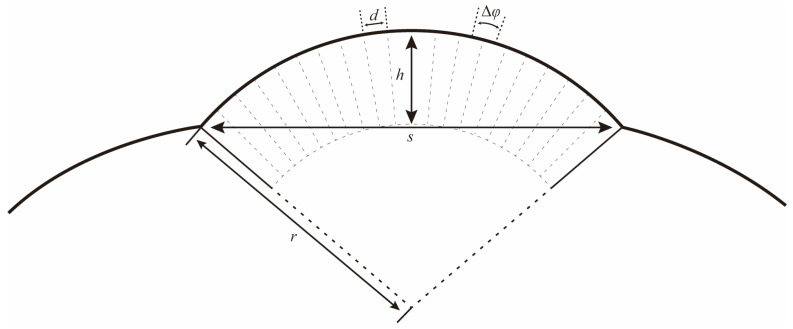
Schematic drawing to illustrate how the local eye radius (*r*) and interommatidial angle (Δ*φ*) were determined. *s*: length of the baseline of the eye; *h*: the longest distance from the curvature to the baseline; and *d*: the facet diameter of an ommatidium.

**Figure 3 insects-15-00532-f003:**
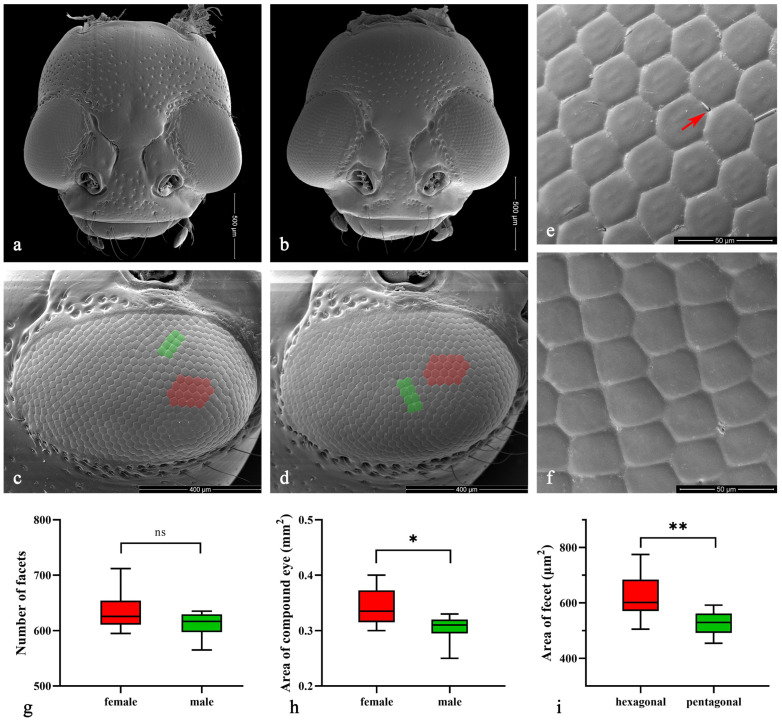
(**a**–**f**) Scanning electron microscopy (SEM) of *Asi. xanthospilota*. (**a**) head of female; (**b**) head of male; (**c**) compound eye of female; (**d**) compound eye of male; (**e**) hexagonal facets; (**f**) pentagonal facets. (**g**–**i**) Measurement of the external morphology of compound eyes. (**g**) Number of facets of male and female (n = 10); (**h**) Area of compound eye of male and female (n = 10); (**i**) Area of each hexagonal and pentagonal facet (n = 20). The red arrow in (**e**) indicates the interfacetal hair; the red areas in (**c**,**d**) indicate parts of the area of hexagonal facets; the green areas indicate parts of the pentagonal facets. A value of *p* < 0.05 was considered statistically significant (n.s., not significant; *, *p* < 0.05; **, *p* < 0.01).

**Figure 4 insects-15-00532-f004:**
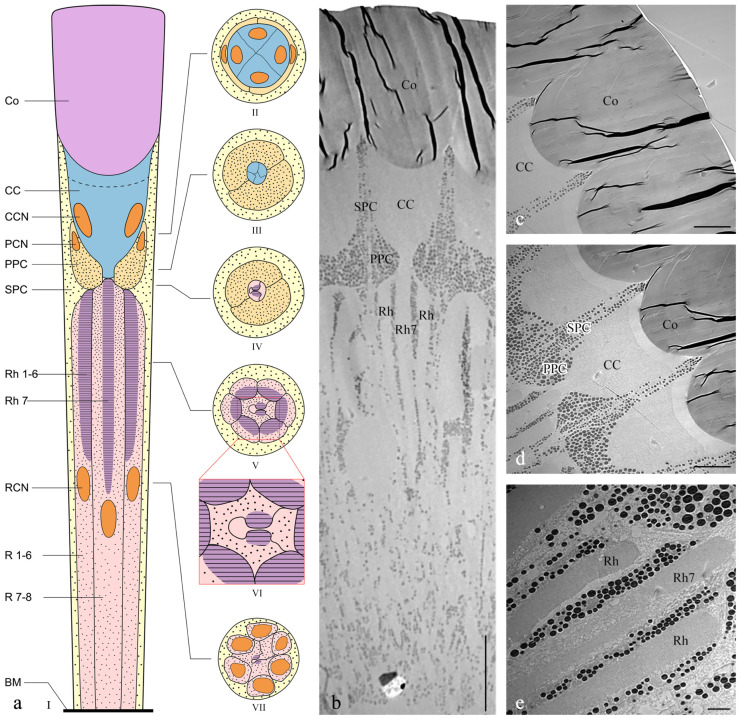
Ultrastructure of ommatidia of *Asi. xanthospilota*. (**a**) Semi-schematic drawings of longitudinal and cross section of an ommatidium; (**b**) TEM micrograph of longitudinal section of ommatidia; (**c**) TEM micrograph of longitudinal section of cornea; (**d**) TEM micrograph of longitudinal section of crystalline cone and primary and secondary pigment cells; (**e**) TEM micrograph of longitudinal section of retinular cells, showing the arrangement of rhabdomeres. Reference figures of the semi-schematic drawings: I—(**b**–**e**); II—Figure 5a,b; III—Figure 5c; IV—Figure 5d; V—Figure 5e; VI—Figure 5f; VII—Figure 5g,h. Abbreviations: Co—cornea; CC—crystalline cone; PPC—primary pigment cell; SPC—secondary pigment cell; Rh—rhabdom; R1–R8—retinular cells; Rh1–Rh7—rhabdomeres; BM—basal membrane; CCN—nuclei of cone cells; PCN—nuclei of primary pigment cells; RCN—nuclei of retinular cells. Scale bar: (**b**) = 20 μm; (**c**,**d**) = 10 μm; (**e**) = 2 μm.

**Figure 5 insects-15-00532-f005:**
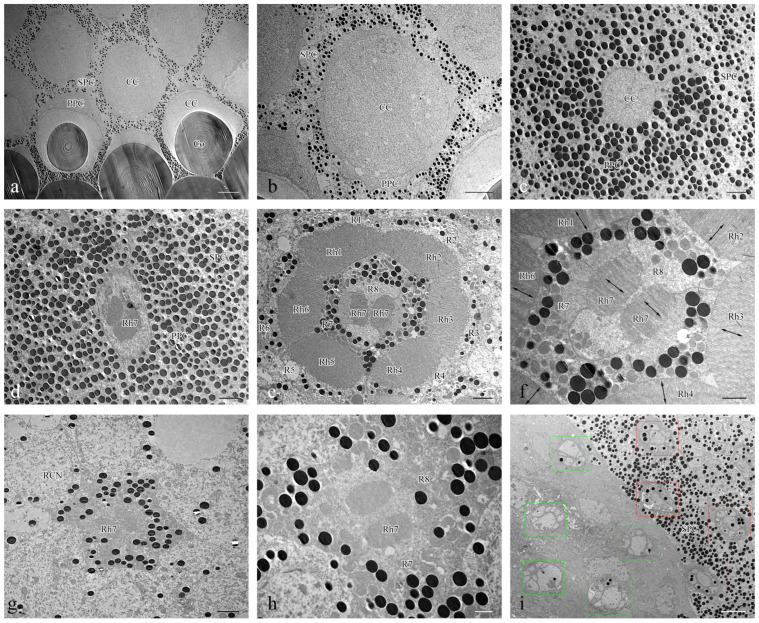
TEM micrographs of cross section at different levels of the compound eye of *Asi. xanthospilota*. (**a**) cross section of cornea and crystalline cone; (**b**) cross section of crystalline cone; (**c**) cross section of crystalline cone and primary pigment cells; (**d**) cross section of distal central rhabdom and primary pigment cells; (**e**) cross section of retinular cells, showing the arrangement of rhabdomeres; (**f**) cross section of central retinular cells, showing the two segments of central rhabdomere attributing by R7 and orientation of microvilli; (**g**) cross section of retinular cells, showing the central rhabdom and nuclei of peripheral retinular cells; (**h**) cross section of central retinular cells, showing the two segments of rhabdomeres generated by R7; (**i**) cross section of the proximal region above and below the basal membrane, showing the arrangements of axons of retinular cells, red rectangular boxes indicate the axon bundles of retinular cells above the basal membrane, green rectangular boxes indicate the axon bundles of retinular cells below the basal membrane. Abbreviations: Co—cornea; CC—crystalline cone; PPC—primary pigment cell; SPC—secondary pigment cell; RCN—nuclei of retinular cells; Rh—rhabdom; R1–R8—retinular cells; Rh1–Rh7—rhabdomeres. Scale bar: (**a**) = 10 μm; (**b**), (**i**) = 5 μm; (**c**–**e**), (**g**) = 2 μm; (**f**,**h**) = 1 μm.

**Figure 6 insects-15-00532-f006:**
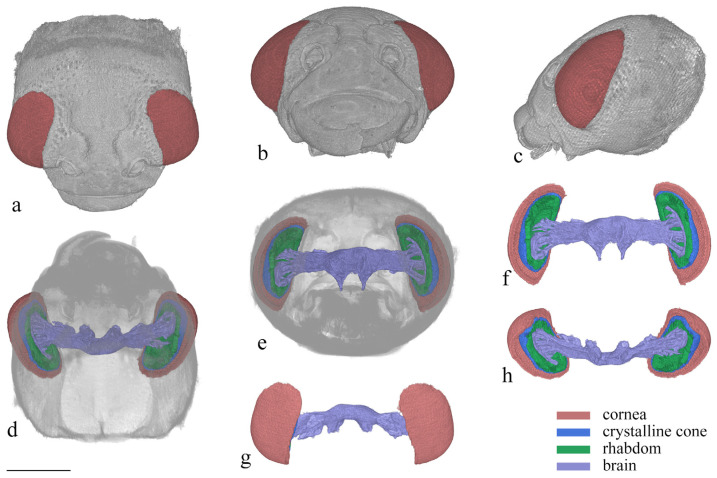
3D reconstruction of the head of *Asi. xanthospilota*. (**a**) dorsal view of head; (**b**) frontal view of head; (**c**) lateral view of head; (**d**) ventral view of head; (**e**) posterior view of head; (**f**) posterior view of compound eyes and brain; (**g**) dorsal view of compound eyes and brain; (**h**) ventral view of compound eyes and brain. Scale bar = 500 μm.

**Figure 7 insects-15-00532-f007:**
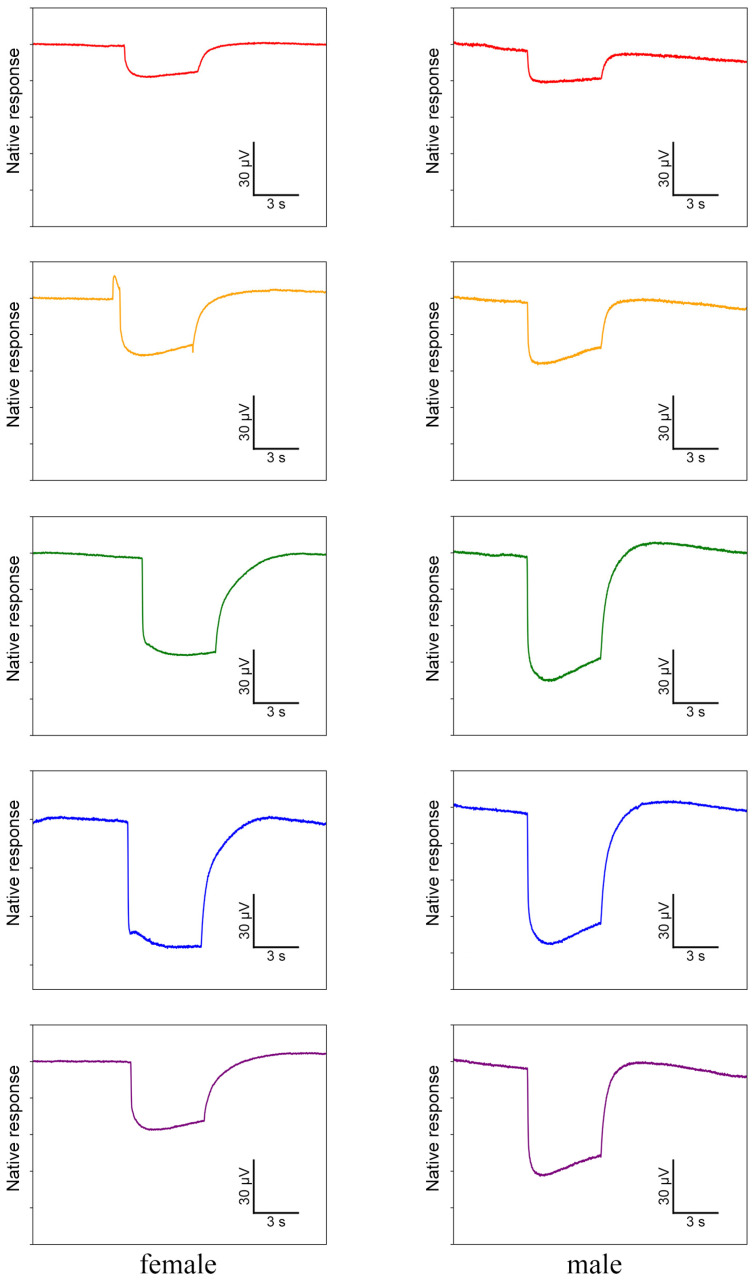
Electrophysiological waveforms of female and male *Asi. xanthospilota* to different light stimuli (red, yellow, green, blue, and ultraviolet from **top** to **bottom**). The waveforms have been denoised using R.

**Figure 8 insects-15-00532-f008:**
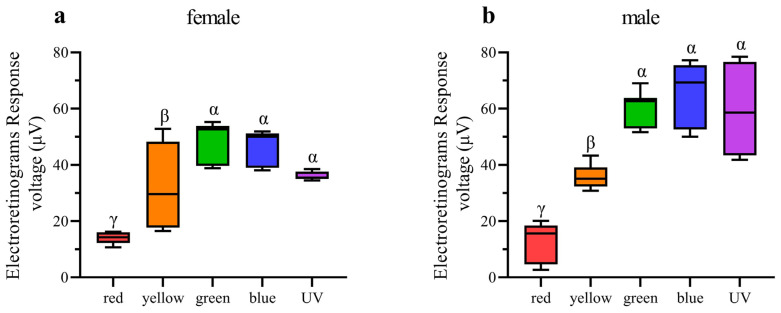
Quantification of ERG voltage responses of female and male *Asi. xanthospilota* exposed to different light stimuli (n = 9). (**a**) female; (**b**) male. Boxplots not sharing the same Greek letter are significantly different at *p*-value < 0.05.

**Figure 9 insects-15-00532-f009:**
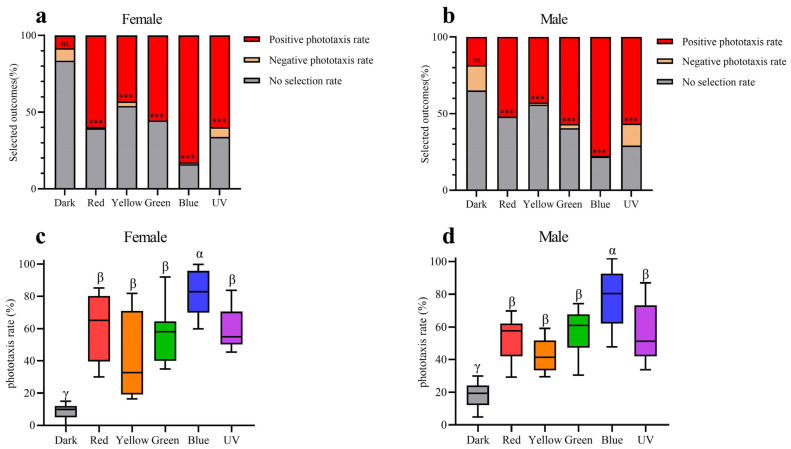
Phototaxis responses of female and male *Asi. xanthospilota* to different light stimuli (n = 9). (**a**) phototaxis response of female different light stimuli; (**b**) phototaxis response of male different light stimuli; (**c**) positive phototaxis rate of female different light stimuli; (**d**) positive phototaxis rate of male different light stimuli. A value of *p* < 0.05 was considered statistically significant (n.s., not significant; ***, *p* < 0.001). Boxplots not sharing the same Greek letter are significantly different at *p*-value < 0.05.

## Data Availability

All analyzed data are available in this paper. However, the raw micro-CT data can be made available upon request to the corresponding author.

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
