# Peer review of "Ultrastructure and Spectral Characteristics of the Compound Eye of *Asiophrida xanthospilota* (Baly, 1881) (Coleoptera, Chrysomelidae)"

_insects, 2024, doi:10.3390/insects15070532_

Round 1

Reviewer 1 Report

Comments and Suggestions for Authors

This manuscript reports the results of structural, physiological, and behavioral analysis of the compound eyes of the chrysomelid beetle species Asiophrida xanthospilota. The study has been diligently designed, performed, and documented. A number of major and minor issues need to be addressed:

1.

“Some molecular studies have revealed that 404

the SW opsin class, which is typically sensitive to blue light, has been lost in some beetle 405

lineages, including Cicindelidae [33], Scarabaeidae [33], Dytiscidae [34,35], Buprestidae 406

[36], Lampyridae [37,38] and Tenebrionidae [39].”

Sharkey et al (2017: Scientific Reports, 7, 8), cited later by the authors, demonstrated the absence of the SW opsin gene subfamily from all Coleoptera. That means also A. xanthospilota lacks SW opsin. One question the authors could easily address by BLAST searches of public databases is whether UV opsin duplications exist in the Chrysomelidae. This knowledge would profoundly benefit the discussion of the physiological and behavioral findings.

2.

How do you explain the differential attraction to blue opsin in the photoresponse test in the absence of SW opsin?

3.

What are the implications of your findings for pest control efforts?

Minor issues:

1.

“an acone of four cone 18

cells”

Whst is an “acone”?

2.

“eight retinular cells along with the open rhabdom”

AN open rhabdom

3.

“After being dehydrated, the samples were 136

dried in a freeze-dryer (Marin Christ) for 12 h. They were then mounted on an Eppendorf 137

tube and scanned using an X-radia scanner (Leica Micro XCT-400) at a magnification of 138

4×. Images were captured at intervals of 5.5 s. 2D image stacks obtained through micro- 139

CT scanning were reconstructed, and different compound eye structures were segmented 140

using Amira software version 6.0.1. The segmented materials were imported into VG 141

Studio Max 3.4.1 for rendering and visualisation.”

Micro-computed tomography and 3D reconstruction were conducted using a single female specimen. So maybe the paragraph needs to be revised into singular?

4.

“resulting in the removal of antennae and prolegs”

Prolegs = gnathal mouthparts?

5.

“Yet, the covering area of the eyes”

surface

6.

“The spatial resolution of the eyes can be estimated 289

using the interommatidial angle (Δφ). The eye radius (r) and Δφ can be determined 290

following the formulas below (Figure 6) [23]: 291

𝑟 =

(𝑠⁄2)

2 + ℎ

2

2ℎ

292

and 293

△ 𝜑 = 𝑑⁄𝑟 294

where srefers to the length of the baseline of the eye, which measures 796 μm; h refers 295

to the longest distance from the curvature to the baseline, measuring 286 μm; and d refers 296

to the facet diameter of an ommatidium, measuring 26.77 μm. “

Move to Materials & Methods

7.

“In ERG testing, both female and male individuals exhibit responses in their 306

compound eyes to all five lights (Figure 7)”

In ERG experiments, both female and male individuals exhibited responses in their 306

compound eyes to all five wavelenths tested (Figure 7)

8.

“3.4. ERGs and phototaxis”

Split into separate paragraphs

9.

“Furthermore, the positive phototaxis rates of all five lights exhibit a 324

statistically significant increase compared to the control group in both genders…”

Genders = sexes

10.

“4.1. Ultrastrcuture of ommatidia”

Ultrastrcuture = Ultrastructure

11.

“The complete absence of R8 has never been reported before in beetles with open rhabdoms.”

The complete absence of the rhabdomere from R8 has never been reported before in beetles with open rhabdoms.

12.

“Insects generally possess three opsin proteins (UV, SW and LW), which typically 400

have maximal sensitivity to ultraviolet (~350 nm), blue (~440 nm), and green (~530 nm) 401

light, respectively, resulting in trichromatic visual systems in insects [29].”

Delete “ in insects” - it is redundant

13.

“According to the result of Sharkey, Fujimoto, Lord, Shin, 444

McKenna, Suvorov, Martin and Bybee [29],”

Sharkey et al [29]

14.

“Considering the loss 446

of R8 in Asi. xanthospilota…”

Considering the lack of rhabdomere formation in the R8 cell of Asi. xanthospilota

15.

“Yer, compared to Asi. xanthospilota, Aga. hygrophila exhibits strong responses to red 453

and yellow light.”

Yer = Yet

Comments on the Quality of English Language

Acceptable

Author Response

Reply to reviewer 1:

Major issue

Comment 1: “Some molecular studies have revealed that 404

the SW opsin class, which is typically sensitive to blue light, has been lost in some beetle 405

lineages, including Cicindelidae [33], Scarabaeidae [33], Dytiscidae [34,35], Buprestidae 406

[36], Lampyridae [37,38] and Tenebrionidae [39].”

Sharkey et al (2017: Scientific Reports, 7, 8), cited later by the authors, demonstrated the absence of the SW opsin gene subfamily from all Coleoptera. That means also A. xanthospilota lacks SW opsin. One question the authors could easily address by BLAST searches of public databases is whether UV opsin duplications exist in the Chrysomelidae. This knowledge would profoundly benefit the discussion of the physiological and behavioral findings.

Reponse 1: The results of Sharkey et al (2017) have shown that UV opsin duplications do exist in species of Chrysomelidae, corresponding to the results of electrophysiological studies. We have changed the expression of the sentence to make it clear. A. xanthospilota exhibited strong sensitivity to blue light, which also can be explained by Sharkey et al ‘s study.

Comment 2: How do you explain the differential attraction to blue opsin in the photoresponse test in the absence of SW opsin?

Response 2: Although lacking the SW opsin, some Chrysomelidae were found to be sensitive to blue light, (Doring et al. 2007, Entomol Gever, 29, 81-95; Du et al. 2023, Zoological Systematics, 48, 193–205; Fan et al. 2023, ZooKeys, 1177, 23–40). This controversy can be explained by the UV opsin duplications (Sharkey et al. 2017). The response to blue light of our species is very likely in the same situation.

Comment 3: What are the implications of your findings for pest control efforts?

Response 3: The results of phototropism experiments indicated that A. xanthospilota exhibited the strongest response to blue light. This could be a potential light source for the control of this species. However, many other insects are also attracted to blue light. It is hard to target this species without killing other insects. Thus, we decided not to discuss it in our paper to avoid misleading.

Minor issues:

Comment 1:  “an acone of four cone 18

cells”

Whst is an “acone”?

Reponse 1: All the dioptric elements of the ommatidium are generated by the four cone cells (Semper cells) including the corneal lens and crystalline cone (eucone or exocone). However, many beetles don’t have a crystalline cone beneath the corneal lens. The space between corneal lens and distal tip of rhabdom is occupied by the cone cells themselves to serve as lightguide. This kind of design is thus called acone. It is a diagnostic of all cucujiforms and higher staphyliniforms. In the case of A. xanthospilota, we didn’t find the eucone or exocone in the TEM images, indicating it has an acone ommatidium.

Comment 2: “eight retinular cells along with the open rhabdom”

AN open rhabdom

Response 2: Thank you for pointing this out. “The” has been changed to “an”.

Comment 3: “After being dehydrated, the samples were 136

dried in a freeze-dryer (Marin Christ) for 12 h. They were then mounted on an Eppendorf 137

tube and scanned using an X-radia scanner (Leica Micro XCT-400) at a magnification of 138

4×. Images were captured at intervals of 5.5 s. 2D image stacks obtained through micro- 139

CT scanning were reconstructed, and different compound eye structures were segmented 140

using Amira software version 6.0.1. The segmented materials were imported into VG 141

Studio Max 3.4.1 for rendering and visualisation.”

Micro-computed tomography and 3D reconstruction were conducted using a single female specimen. So maybe the paragraph needs to be revised into singular?

Response 3: Thank you for pointing this out. This comment is adopted in the revised manuscript.

Comment 4: “resulting in the removal of antennae and prolegs”

Prolegs = gnathal mouthparts?

Resonse 4: “Prolegs” refers to the forelegs. To keep the head intact, the head and prothorax were cut down together, and then the antennae and forelegs on prothorax were removed.

Comment 5: “Yet, the covering area of the eyes”

surface

Response 5: We agree with the comment. This comment is adopted in the revised manuscript.

Comment 6: “The spatial resolution of the eyes can be estimated 289

using the interommatidial angle (Δφ). The eye radius (r) and Δφ can be determined 290

following the formulas below (Figure 6) [23]: 291

? =

(?⁄2)

2 + ℎ

2

2ℎ

292

and 293

△ ? = ?⁄? 294

where srefers to the length of the baseline of the eye, which measures 796 μm; h refers 295

to the longest distance from the curvature to the baseline, measuring 286 μm; and d refers 296

to the facet diameter of an ommatidium, measuring 26.77 μm. “

Move to Materials & Methods

Response 6: We agree with the comment. This comment is adopted in the revised manuscript. The formulas have been moved to Materials and Methods.

Comment 7: “In ERG testing, both female and male individuals exhibit responses in their 306

compound eyes to all five lights (Figure 7)”

In ERG experiments, both female and male individuals exhibited responses in their 306

compound eyes to all five wavelenths tested (Figure 7)

Response 7: Thank you for pointing this out. The tense has been revised in the manuscript.

Comment 8: “3.4. ERGs and phototaxis”

Split into separate paragraphs

Response 8: We agree with the comment. The results of ERG and phototaxis have been spilt into separate paragraphs.

Comment 9: “Furthermore, the positive phototaxis rates of all five lights exhibit a 324

statistically significant increase compared to the control group in both genders…”

Genders = sexes

Response 9: Thank you for pointing this out. “Genders” has been changed to “sexes” in the manuscript.

Comment 10: “4.1. Ultrastrcuture of ommatidia”

Ultrastrcuture = Ultrastructure

Response 10: Thank you for pointing this out. The misspelling issue of “Ultrastrcuture” has been corrected.

Comment 11: “The complete absence of R8 has never been reported before in beetles with open rhabdoms.”

The complete absence of the rhabdomere from R8 has never been reported before in beetles with open rhabdoms.

Response 11: The retinular cell was abbreviated with “RC” and rhabdomere was abbreviated with “R” in the original manuscript. After consulting references, we agree with the comment and use “R” to represent retinular cell and use “Rh” to represent rhabdomere.

Comment 12: “Insects generally possess three opsin proteins (UV, SW and LW), which typically 400

have maximal sensitivity to ultraviolet (~350 nm), blue (~440 nm), and green (~530 nm) 401

light, respectively, resulting in trichromatic visual systems in insects [29].”

 Delete “ in insects” - it is redundant

Response 12: Thank you for pointing it out. The redundant words “in insect” have been removed.

Comment 13: “According to the result of Sharkey, Fujimoto, Lord, Shin, 444

McKenna, Suvorov, Martin and Bybee [29],”

Sharkey et al [29]

Response 13: Thank you for pointing it out. The citation format has been corrected.

Comment 14: “Considering the loss 446

of R8 in Asi. xanthospilota…”Considering the lack of rhabdomere formation in the R8 cell of Asi. xanthospilota

Response 14: Thank you for pointing it out. The sentence has been revised as required.

Comment 15: “Yer, compared to Asi. xanthospilota, Aga. hygrophila exhibits strong responses to red 453

and yellow light.”

Yer = Yet

Response 15: Thank you for pointing it out. The misspelling issue has been corrected.

Reviewer 2 Report

Comments and Suggestions for Authors

MS insect3050393

Authors analysed the compound eyes in the beetle Asiophrida xanthospilota employing a range of techniques, including microscopy, micro-CT and physiological methodology.

The results are a contribution to the knowledge of morphology and function of eyes in Coleoptera. The experimental set up is well designed and the methods applied are appropriate.

The manuscript is suitable for publication following the revisions as outlined below.

Introduction

1. In L78, 84,85 and throughout the text, authors should pay attention to the taxonomic roles for genus abbreviation, revise A. hygrophila, A. xanthospilota, C. coggygria and L99

2. Micro CT is an invaluable, exploratory methodology that offers an alternative to invasive and time-consuming techniques for investigating morphology. It is therefore recommended that the background or discussion section be improved by including a paragraph outlining recent micro-CT studies on compound eyes such as

Currea, John Paul, et al. "Measuring compound eye optics with microscope and microCT images." Communications biology 6.1 (2023): 246.

Romell, Jenny, et al. "Laboratory phase‐contrast nanotomography of unstained Bombus terrestris compound eyes." Journal of Microscopy 283.1 (2021): 29-40.

Giglio, A., Vommaro, M. L., Agostino, R. G., Lo, L. K., & Donato, S. (2022). Exploring compound eyes in adults of four coleopteran species using synchrotron X-ray phase-contrast microtomography (SR-PhC micro-CT). Life, 12(5), 741.

Tichit, P., Zhou, T., Kjer, H. M., Dahl, V. A., Dahl, A. B., & Baird, E. (2022). InSegtCone: interactive segmentation of crystalline cones in compound eyes. BMC zoology, 7(1), 10.

Paukner, D., Wildenberg, G. A., Badalamente, G. S., Littlewood, P. B., Kronforst, M. R., Palmer, S. E., & Kasthuri, N. (2024). Synchrotron‐source micro‐x‐ray computed tomography for examining butterfly eyes. Ecology and Evolution, 14(4), e11137.

Muinde, J., Zhang, T. H., Liang, Z. L., Liu, S. P., Kioko, E., Huang, Z. Z., & Ge, S. Q. (2024). Functional Anatomy of Split Compound Eyes of the Whirligig Beetles Dineutus mellyi (Coleoptera: Gyrinidae). Insects, 15(2), 122.

M&M

In general, some sentence should be revised.

3. L98-99 add geographical coordinates

4. L116 insert M and pH of buffer

5. L120-121 it should be indicated the buffer used to wash samples and  revised the sentence “dehydrated by in a graded series….”

6. L120-123 Dehydration in ethanol followed by immersion in acetone is not useful for Spurr infiltration, it is better to dehydrate in a series of acetone.

7. L116, 119,124 The trademark of chemicals should be indicated

8. L136 revise “Then, the samples were…..”

9. L163 the sentence should be revised to specify type and manufacturer of instrument used as light source.  Revise “ …and ultraviolet (365–400 nm) wavelengths were tested …”

10. L182 this sentence states an information about p-value and number of text absent in figures.

Results

11. This section should be improved inserting in brackets p-value and number of tests were statistical differences are indicated

12. L194-195 the area of hexagonal and pentagonal facets should be specified in the text and/or indicated in fig2c and d using arrowheads or colours

13. L261-264 lack the figure

14. L273 revise “Three distinct layers are observed in the reconstructed 3D images, including the corneal layer, the….”

15. L276-279 Structures would have been visible in virtual section ad 3D reconstruction if a fixative such as glutaraldehyde or similar had been used instead of ethanol.

16. Fig7  the x- and y-axis legend should be enlarged as it is currently too small to read.

17. L180, 307, 322 “lights” meaning “light frequency”, the text should be revised to indicate, the frequency that correspond to the colours indicated

18. L306-308 insert the figure

20. 19. L310-316 the text should be improved adding the statistical text used, p-value and number of responses

21. Fig 8 the reference text should be revised, a and b labels hare used to indicate both  figures and statistical differences. Moreover, a sentence to indicate statistical difference in the boxes should be used , for ex “Boxplots not sharing the same letter are significantly different at p-value < xxx” .

22. L330-331 The figure showing phototaxis assay result should be renamed and corrected in the text. Moreover, see the comment of fig.8 and revise according to.

Author Response

Reply to reviewer 2:

Comment 1: In L78, 84,85 and throughout the text, authors should pay attention to the taxonomic roles for genus abbreviation, revise A. hygrophila, A. xanthospilota, C. coggygria and L99

Reponse 1: Normally, the genus name is abbreviated with the first character and a period in the abbreviated scientific names. However, in the case of our manuscript, both Asiophrida xanthospilota and Agasicles hygrophila have their genus name initiated with ‘A’. Thus, to distinguish the two genera, we have to get more characters involved. That’s why the names were abbreviated as Asi. xanthospilota and Aga. hygrophila. The case of Cotinus coggygria and Carabus nemoralis is in the same situation.

Comment 2: Micro CT is an invaluable, exploratory methodology that offers an alternative to invasive and time-consuming techniques for investigating morphology. It is therefore recommended that the background or discussion section be improved by including a paragraph outlining recent micro-CT studies on compound eyes such as

Currea, John Paul, et al. "Measuring compound eye optics with microscope and microCT images." Communications biology 6.1 (2023): 246.

Romell, Jenny, et al. "Laboratory phase‐contrast nanotomography of unstained Bombus terrestris compound eyes." Journal of Microscopy 283.1 (2021): 29-40.

Giglio, A., Vommaro, M. L., Agostino, R. G., Lo, L. K., & Donato, S. (2022). Exploring compound eyes in adults of four coleopteran species using synchrotron X-ray phase-contrast microtomography (SR-PhC micro-CT). Life12(5), 741.

Tichit, P., Zhou, T., Kjer, H. M., Dahl, V. A., Dahl, A. B., & Baird, E. (2022). InSegtCone: interactive segmentation of crystalline cones in compound eyes. BMC zoology7(1), 10.

Paukner, D., Wildenberg, G. A., Badalamente, G. S., Littlewood, P. B., Kronforst, M. R., Palmer, S. E., & Kasthuri, N. (2024). Synchrotron‐source micro‐x‐ray computed tomography for examining butterfly eyes. Ecology and Evolution14(4), e11137.

Muinde, J., Zhang, T. H., Liang, Z. L., Liu, S. P., Kioko, E., Huang, Z. Z., & Ge, S. Q. (2024). Functional Anatomy of Split Compound Eyes of the Whirligig Beetles Dineutus mellyi (Coleoptera: Gyrinidae). Insects15(2), 122.

Response 2: We agree with the comment. The background of recent micro-CT studies has been added in the introduction.

Comment 3: L98-99 add geographical coordinates

Response 3: We agree with the comment. Geographical coordinate has been added.

Comment 4: L116 insert M and pH of buffer

Response 4: We agree with the comment. M and pH of the buffer have been added.

Comment 5: L120-121 it should be indicated the buffer used to wash samples and  revised the sentence “dehydrated by in a graded series….”

Response 5: We used ddH2O to wash the samples, which has been indicated in the revised manuscript. The sentence has been revised as advised.

Comment 6: L120-123 Dehydration in ethanol followed by immersion in acetone is not useful for Spurr infiltration, it is better to dehydrate in a series of acetone.

Response 6: According to our colleague in charge of TEM sample preparation, both ethanol and acetone are excellent dehydrating agents. However, the extraction of cell material caused by ethanol is smaller than that of acetone. So we used ethanol as the dehydrating agent.

Comment 7: L116, 119,124 The trademark of chemicals should be indicated

Response 7: We agree with the comment. The trademark of chemicals has been indicated.

Comment 8: L136 revise “Then, the samples were…..”

Response 8: Thank you for pointing this out. The sentence has been revised.

Comment 9: L163 the sentence should be revised to specify type and manufacturer of instrument used as light source.  Revise “ …and ultraviolet (365–400 nm) wavelengths were tested …”

Response 9: We accept the comment. The light sources were assembled by ourselves with LED lamp bead. The sentence has been revised.

Comment 10: L182 this sentence states an information about p-value and number of text absent in figures.

Response 10: Thank you for pointing it out. The p-value and number of tests had been added in legends.

Comment 11: This section should be improved inserting in brackets p-value and number of tests were statistical differences are indicated

Response 11: Thank you for pointing it out. The p-value and number of tests had been added in brackets.

Comment 12: L194-195 the area of hexagonal and pentagonal facets should be specified in the text and/or indicated in fig2c and d using arrowheads or colours

Response 12: We accept the comment. The area of hexagonal and pentagonal facets has been indicated with coloured masks.

Comment 13: L261-264 lack the figure

Response 13: We accept the comment. The orientation of microvilli is shown in Figure 5f. The citation has been added.

Comment 14: L273 revise “Three distinct layers are observed in the reconstructed 3D images, including the corneal layer, the….”

Response 14: Thank you for pointing it out. The sentence has been revised as required.

Comment 15: L276-279 Structures would have been visible in virtual section ad 3D reconstruction if a fixative such as glutaraldehyde or similar had been used instead of ethanol.

Response 15: Ethanol has always been the fixative and dehydrating agent used in our micro-CT protocols and we haven’t used glutaraldehyde. Glutaraldehyde could be a good fixative and dehydrating agent; we could try it in future studies and compare it with ethanol.

Comment 16: Fig7  the x- and y-axis legend should be enlarged as it is currently too small to read.

Response 16: We agree with the comment. The legend of the figures has been revised to enlarged the text.

Comment 17: L180, 307, 322 “lights” meaning “light frequency”, the text should be revised to indicate, the frequency that correspond to the colours indicated

Response 17: We agree with the comment. The word “lights” has been changed to “light wavelengths”. Most pubulcations regarding insect’s color vision use “wavelength” instead of “frequency” to define different light colors. Thus, we use the word “wavelength” throughout the manuscript.

Comment 18: L306-308 insert the figure

Response 18: We agree with the comment. The figures have been cited in the text.

Comment 19-20: L310-316 the text should be improved adding the statistical text used, p-value and number of responses

Response 19-20: We agree with the comment. The p-value and number of tests had been added in brackets in legends of figure 8.

Comment 21: Fig 8 the reference text should be revised, a and b labels hare used to indicate both  figures and statistical differences. Moreover, a sentence to indicate statistical difference in the boxes should be used , for ex “Boxplots not sharing the same letter are significantly different at p-value < xxx” .

Response 21: We agree with the comment. We have changed statistical differences with Greek letter in Figure 8-9 to avoiding misunderstanding. The sentence to indicate statistical difference in the boxes has been added in the legends of figures 8-9.

Comment 22: L330-331 The figure showing phototaxis assay result should be renamed and corrected in the text. Moreover, see the comment of fig.8 and revise according to.

Response 22: Thank you for pointing it out. The figure number has been corrected to “Figure 9”.

Reviewer 3 Report

Comments and Suggestions for Authors

­Liang et al used EM and microCT to study the structure of the compound eyes as well as individual ommatidia of flea beetles Asiophrida xanthospilota.  In addition, the authors used electroretinogram to measure the beetles’ spectral sensitivity. Although the aim to understand how the beetles eyes sample the environment is interesting by itself, I found many of the stated results are less than convincing given the data shown in the manuscript. I will list my main concerns below. I suggest the authors should either present their data in a more lucid and specific fashion, or reconsider the conclusion of this study.

1.      One of the main results stated is that the flea beetle has apposition eyes. Why is this interesting or novel given that it’s expected for the leaf beetle family?  Is there any evidence to expect otherwise?

2.      Line 67, the clear zone does not allow one ommatidium to effective collect light from “hundreds of facets” – that would completely blur the image. I appreciate a pedagogical introduction but please make sure all facts are correct.

3.      Little was said about the EM and microCT specifications and data quality – what are the experimental parameters used in the scans? What are the voxel size or resolutions of the data? Why is it necessary to use SEM to measure the surface of the compound eye (does the microCT not have sufficient resolution) ?

4.      Figure 1 is of minimal value. If keep, please add a reference. Instead, I suggest add more informative plots in figures. For instance, the measurements in and around line 189 should be plotted.

5.      Figure 3, I found this result unclear: most results are stated based on panel a, which is a schematic drawing. I had a very difficult time to see these structure in the gray scale images in the other panels. If they’re there, please add clear and sufficient annotations. At this point, as far as I’m concerned, most of the observations stated in the text are not supported by data.

6.      What are the dark disks in these EM images? Why does the contrast vary so much between , say, panel b and e?

7.      What’s the origin of the laminated structure in the cornea? Are the authors certain this is not a defect caused by the sample preparation?  Such a structure might well alter the light diffraction in the cornea.

8.      Figure 4e, how could the authors conclude that those 2 central rhabdomeres are both from R7? R8 usually comes below R7. Has the authors checked more proximal cross section to rule out the possibility of an R8 rhabdomere ?

9.      Line 153, lux is the unit of illuminance, not intensity.

10.  The main concern with the electroretinogram measurement is the intensity of the light. Illuminance (lux) is measured based on human vision, not necessarily appropriate for beetles. A large electrical response doesn’t necessarily guarantee higher sensitivity to a particular wavelength since the light intensity are not “normalized” to what a beetle normally see in a naturalistic environment.

11.  No reference in the title 

Comments on the Quality of English Language

ok

Author Response

Reply to reviewer 3:

Comment 1: One of the main results stated is that the flea beetle has apposition eyes. Why is this interesting or novel given that it’s expected for the leaf beetle family?  Is there any evidence to expect otherwise?

Response 1: We didn’t consider the apposition eye of Asiophrida xanthospilota an interesting or novel result. But the type of eyes is an indispensable trait to describe the ultrastructure of insect compound eye. That is why we described it in results.

Comment 2: Line 67, the clear zone does not allow one ommatidium to effective collect light from “hundreds of facets” – that would completely blur the image. I appreciate a pedagogical introduction but please make sure all facts are correct.

Response 2: Thank you for pointing it out. The word “hundreds of” has been changed to “multiple” to ensure the preciseness.

Comment 3: Little was said about the EM and microCT specifications and data quality – what are the experimental parameters used in the scans? What are the voxel size or resolutions of the data?

Why is it necessary to use SEM to measure the surface of the compound eye (does the microCT not have sufficient resolution) ?

Response 3: The image size of the reconstructed images as well as pixel size has been indicated. We used SEM to measure the surface because it has a better resolution than microCT and can provide more details about the surface of the eyes.

Comment 4: Figure 1 is of minimal value. If keep, please add a reference. Instead, I suggest add more informative plots in figures. For instance, the measurements in and around line 189 should be plotted.

Response 4: Figure 1 was taken by one of the authors, Dr. Zhengzhong Huang. The photographer has been indicated. We believe this figure can leave an intuitive impression on readers about the research object of our study. The measurement of the external morphology of the eye has been plotted and added to Figure 3.

Comment 5: Figure 3, I found this result unclear: most results are stated based on panel a, which is a schematic drawing. I had a very difficult time to see these structure in the gray scale images in the other panels. If they’re there, please add clear and sufficient annotations. At this point, as far as I’m concerned, most of the observations stated in the text are not supported by data.

Response 5: Panel a in Figure 3 (Figure 4 in the revised manuscript) was drawn based on TEM images in Figures 3 and 4 (Figures 4 and 5 in the revised manuscript) and some other TEM images not shown in the Figures, and we also referred to other paper about the compound eye of leaf beetle for comparison. The drawing is a conventional way to help explain the ultrastructure of ommatidia. We added the necessary annotation of the structure with abbreviation on the figures. Reference figures of the drawings have been indicated in the legends of Figure 4 to make it easier to understand the structure of the ommatidia.

Comment 6: What are the dark disks in these EM images? Why does the contrast vary so much between , say, panel b and e?

Response 6: During the preparation of TEM samples, the slices float on the surface of the water due to the surface tension after slicing. The cornea is composed of hard composition, so wrinkles will be formed due to incomplete spreading, and the wrinkles will greatly reduce the electron transmitting rate under the electron microscope, causing black-banded areas in the image.

The imaging software will automatically adjust the contrast according to the grey scale of the image. When the darker areas of the picture (e.g. pigment particles, folds, impurities) are more predominant, the rest of the area will be brighter; conversely, if the lighter areas of the picture (e.g. holes) are more predominant, the picture as a whole will appear darker. That’s why the contrast varies between images.

Comment 7: What’s the origin of the laminated structure in the cornea? Are the authors certain this is not a defect caused by the sample preparation?  Such a structure might well alter the light diffraction in the cornea.

Response 7: The laminated structure of the cornea is formed during corneal formation. The cornea is secreted layer by layer by the Semper’s cells situated below the cornea. This structure is common in the cornea of insect’s compound eye, so we are confident that is it not defect caused by the sample preparation.

Comment 8: Figure 4e, how could the authors conclude that those 2 central rhabdomeres are both from R7? R8 usually comes below R7. Has the authors checked more proximal cross section to rule out the possibility of an R8 rhabdomere ?

Response 8: This is a very constructive question. There are mainly three reasons why we believe bot the two central rhabdomeres belong to R7. First, the symmetry of the two rhabdomeres and the two central cells indicated that it is from the same cells, either R7 or R8. Second, Figure 5d shows the distal end of the central retinular cells, where we can see that only one cell body is present at this level (i.e. R7), together with two rhabdomeres attached to each other, indicating that they are generated by R7. Third, our previous paper [Fan, W.-L.; Liu, X.-K.; Zhang, T.-H.; Liang, Z.-L.; Jiang, L.; Zong, L.; Li, C.-Q.; Du, Z.; Liu, H.-Y.; Yang, Y.-X. The morphology and spectral characteristics of the compound eye of Agasicles hygrophila (Selman & Vogt, 1971) (Coleoptera, Chrysomelidae, Galerucinae, Alticini). ZooKeys 2023, 1177, 23–40.] studied the ultrastructure of ommatidia of the other flea beetles Agasicles hygrophila, which revealed that the central rhabdom of this species consists of two rhabomeric segments from one cell and another segment from the other cells. Comparing the arrangement of the rhbdomeric segments of the two species, we can conclude that both the two rhbdomeric segments of Asi. xanthospilota belongs to R7. In our previous TEM slices, unfortunately we didn’t have slice of proximal sections.

To rule out the possibility of an R8 rhabdomere in the proximal end, we prepared some new samples to get the slices of proximal part of the ommatida recently and added some new TEM images in Figure 5. In Figures 5g-h, we can see that at the proximal level nuclei of peripheral retinular cells emerge and peripheral rhabdomeres disappear, the two central rhabdomeres still exist as a pair but no sign of the R8 rhabdomere were found. We have multiple similar images showing the same pattern, so we are confident that the R8 rhabdomere doesn’t exist.

Comment 9: Line 153, lux is the unit of illuminance, not intensity.

Response 9: Thank you for pointing it out. “intensity” has been changed to “illuminance”.

Comment 10: The main concern with the electroretinogram measurement is the intensity of the light. Illuminance (lux) is measured based on human vision, not necessarily appropriate for beetles. A large electrical response doesn’t necessarily guarantee higher sensitivity to a particular wavelength since the light intensity are not “normalized” to what a beetle normally see in a naturalistic environment.

Response 10: Luminous intensity is a quantity directly related to power of light, which describes the light source itself, while illuminance represents the quantity of light falling onto a given surface area, which is related to distance, area, and other factors. For example, for the light sources we use for electrophysiology and behaviour, such as blue light, the intensity of the light is constant regardless of the distance from the compound eye, but the illuminance becomes weaker as the distance increases. We mixed up light intensity and illuminance in our manuscript. What we controlled was illuminance instead of intensity. In our study, We intended to control the consistency of the illuminance and investigate the sensitivity of Asi. xanthospilota to different light wavelengths with the same illuminance. So we believe it is necessary to keep the illuminance consistent instead of “normalized” it.

Comment 11: No reference in the title

Response 11: We are sorry we don’t understand what this comment refers to. Can you make it more specific?